# Rapid bacteriophage quantification with a particle size analyzer combined with polarization intensity differential scattering (PIDS) detector

Amanda Carroll-Portillo[1]*, Cody A Braun[2], Edgar Martinez[3], Henry C Lin[1,4]*

1 Division of Gastroenterology and Hepatology, University of New Mexico, Albuquerque, New Mexico, United States of America, 2 Biomedical Research Institute of New Mexico, Albuquerque, New Mexico, United States of America, 3 Beckman Coulter, Inc., Miami Campus, Miami, Florida, United States of America, 4 Medicine Service, New Mexico VA Health Care System, Albuquerque, New Mexico, United States of America

* acarroll-Portillo@salud.unm.edu (ACP); helin@salud.unm.edu (HCL)

## Abstract

The study of bacteriophages (phages) and effects on their microenvironments expanded exponentially within the last decade. While there are multiple described methods for phage quantitation, there is still a need for a rapid, label-free method. To this end, we established a procedure for rapid phage quantitation through novel use of a particle size analyzer with Polarization Intensity Differential (PIDS) technology and eliminated the need for labels or knowledge of bacterial host. We validated the procedure and analysis method, termed PhageFOTO (Fast Optical Tallying of Objects) using several physiologically different phages ranging from ~6 nm capsid width (*Inoviridae*) to ~90 nm capsid width (*Caudoviricetes*). PhageFOTO demonstrated 89±4.3%, 98±1.7%, and 94±2.7% accuracy for quantitating PhiX, M13, and T4 phages/mL respectively as compared to the gold standard plaque assay with limit of detection for particle concentration occurring around $10^7$ phages/mL. PhageFOTO proved to be a novel, rapid, label free method for phage counting that does not rely on knowledge of the bacterial host presenting unique capability for quantitation of phage samples.

## Introduction

Bacteriophages (phages) are ubiquitous, bacteria-targeting viruses with profound impact on their microenvironments. Phage effects increasingly are found to influence health, medicine, and the food and drug industries [1]. An essential part of this research requires quantitation of viral particles within any given sample, an ongoing challenge given their small size and lack of consensus genetic sequence (i.e., 16S or 18S). Currently, there are a variety quantitation methods of varying ease and expense as reviewed partially by Acs N, *et al.* [2]. Of these methods, Transmission

**Data availability statement:** All relevant data are within the paper and its Supporting Information files.

**Funding:** This work was supported by the Winkler Bacterial Overgrowth Research Fund (BRINM 217). The funders had no role in study design, data collection and analysis, decision to publish, or preparation of the manuscript.

**Competing interests:** I have read the journal's policy and the authors of this manuscript have the following competing interests: E. Martinez is an employee of Beckman Instruments. All PSA instrumentation was purchased from Beckman Instruments. Authors A. Carroll-Portillo, C. Braun, and H.C. Lin have no competing financial interests and received no financial support from Beckman Instruments for this study. Beckman Instruments had no editorial control over the content of this paper.

Electron Microscopy (TEM) is the most accurate involving direct visualization of the morphology and number of phages within a sample. However, this method is time consuming, requires expensive equipment and experience with the technique, and has a high level of bias as to what is counted as a phage. Other methods that are used with more frequency include the plaque assay or fluorescent labeling for either epifluorescent microscopy or flow cytometry. The plaque assay involves enumerating phage-derived plaques, or plaque forming units (PFU), on a plated lawn of that phage's specific bacterial host thus determining the active phage numbers in a sample. Counts from PFU of a known dilution of phage stock solution allows for back calculation of the phage titer in the original sample. While the gold standard for phage quantitation, this method is insufficient when working with an environmental sample (multiple types of bacterial hosts), a bacterial host that is difficult to grow (e.g., anaerobic bacteria), or when plaques are minuscule or hazy in appearance making counting difficult if not impossible.

As such, for environmental sampling, where microbial communities contain phages for a variety of bacterial hosts, or other difficult samples fluorescence-based methods of epifluorescent microscopy or flow cytometry are used for quantitation [2–6]. These methods both require labeling of phage nucleic acid or proteins with fluorescent dyes. Epifluorescent microscopy (EFM) involves counting fluorescent phages labeled with an intercalating dye (e.g., SYBR Green) applied to 0.02 μm filters with vacuum. Error in EFM is introduced through non-uniform labeling of nucleic acid, non-phage particle labeling (i.e., exosomes containing nucleic acid), or resolution limitations of the microscope itself (more than one phage per "spot").

The alternative to EFM is flow cytometry. This method also uses labeling of phages to provide quantitative results [3,4,7–9] with either nucleic acid intercalators or fluorophores conjugated to or expressed by the phages of interest [10,11]. Quantitation of phages with standard flow cytometry works at the edge of the detection limit of the instrument. While rapid, sample handling greatly effects results [3,6] with stringent need for use of appropriate controls [5,9,12].

With these caveats in mind for current protocols, we developed a novel method of phage quantitation with a particle size analyzer for the first time to generate rapid results (within an hour) without need of fluorescent labeling or knowledge of bacterial host. Particle size analyzers (PSA) have long been used for characterization of small particles in the explosives and air quality fields to detect aerosolized particles within a sample [13,14]. Nanochemistry applications have also employed particle counters to quantify a variety of synthesized particles (i.e., quantum dots) within solutions [15,16]. A newer PSA with advanced Polarization Intensity Differential Scattering (PIDS) technology (LS 13320XR) detects particle size distributions within the range phages are found introducing a new application for this instrumentation.

All particles scatter light, but smaller particles do so at higher angles making their size distribution more difficult to measure. By employing PIDS, which utilizes detection of scattering due to three different wavelengths at two different polarizations, in combination with Mie scattering theory for spherical particles [17], we are able to detect particles in the nanometer size distribution (at a lower threshold of ~10nm).

This ability moves detection of PSAs into the range where even the smallest phages could, in theory, be detectable by these instruments. In this method, we describe use of the PSA with PIDS technology for characterizing and quantifying phages within individual phage preparations. We describe use of the PSA across a range of phage sizes and shapes. As this method offers rapid results through an optical method for detecting particles, we have termed this procedure for phage detection PhageFast Optical Tallying of Objects, or PhageFOTO.

## Methods

### Instrumentation

The particle size analyzer, LS 13 320 XR with Polarization Intensity Differential Scattering (PIDS) technology and the universal liquid module (ULM), was purchased from Beckman Coulter (Indianapolis, IN, USA). A refractometer (J457 Automatic Refractometer) was purchased from Rudolph Research Analytical (Hackettstown, NJ) to determine the refractive index (RI) of each phage-containing solution prior to sampling on the PSA. The RI is a necessary software input to allow for accurate particle detection by the PIDS. MilliQ water was used as the carrier fluid for all samples.

### Instrument settings

The PSA software settings for all individual phage samples were as follows:

- Procedure tab – Runs: 4, Run Time: 120 seconds, Sample Loading: 10%, Target range: ±2%

- Module settings tab – Debubble: checked, Circulation speed: 10%, Rinse cycles: 3, Drain time: 10 seconds

- Advanced settings tab – ADAPT mode selected

- Sample properties tab – Enable PIDS: checked, Always include PIDS data when solving: checked, Manage materials: phage to be tested selected (Refractive indices for all samples were manually entered into the software. For individual phages tested, the refractive index for SM Buffer (1.3351) was used), Carrier fluid: water (RI = 1.3330)

SM Buffer was selected as the RI setting for all individual phage protocols as this was the buffer phage-containing solutions were exchanged to prior to sampling (see "Bacteriophage" section).

### Bacteriophage

Individual phages: T4, PhiX 174 (ΦX), and M13 coliphages as well as their *Escherichia coli* host (*E. coli*, strain B) were purchased from ATCC (Manassas, VA). All individual phages were propagated on *E. coli,* Strain B as previously described [18]. Briefly, 1mL of $1 \times 10^8$ plaque forming units (PFU)/mL phage stock was added to 10 mL mid-log bacterial culture in Luria-Bertani broth (Miller) (MilliporeSigma, St. Louis, MO) and allowed to continue shaking at 37°C for an additional 3–4 hours to allow for phage propagation and bacterial lysis. Bacterial remnants were pelleted with centrifugation in a swinging bucket rotor at 4000 rpm for 15 min. Supernatants were filter sterilized with a 0.45 μm PTFE syringe filter and concentrated with Amicon Ultra 10000 molecular weight cutoff (10K MWCO) centrifugal filters (MilliporeSigma, St. Louis, MO) in a swinging bucket rotor at 4000 rpm for 15 min at room temperature (or until volume was reduced to 200–500 μL). Concentrated volumes were then brought back up to 5 mL in SM Buffer (100mM NaCl, 8mM $MgSO_4 \cdot 7H_2O$, 50mM Tris-Cl, pH 7.5) and either applied to the PSA for analysis or diluted for plaque overlay assay. For sensitivity assays, Cesium chloride (CsCl) purified phages were used. This was achieved by layering filter sterilized, concentrated T4 onto a gradient consisting of 3 mL layers of 1.7, 1.5, and 1.3 density CsCl resuspended in CsCl buffer (50mM Tris pH 7.6; 100 mM $MgCl_2$). Samples were centrifuged for 2 hours at 100,000 *x g* at 4°C. Phage bands were withdrawn with a glass pipette and dialyzed in 1 L volume of SM Buffer with dialysis cassettes (10K MWCO, Thermo-Fisher, Waltham, MA) overnight at 4°C.

 

## Phage detection and quantitation

**PSA detection:** Phage containing samples were read on the PSA in triplicate three separate times (Fig 1). Each phage sample was subjected to one or two consecutive reads with 4 runs/read. Results from individual phages (T4, ΦX, and M13) were used for instrument validation. Volume percent data, mean peak size, obscuration numbers (PIDS (%)), and sample volume were input into the PhageFOTO macro (green highlighted fields in S1 File) to calculate estimated particle number/mL for each read.

**Plaque overlay assay quantification:** To quantify phages in individual phage solutions, plaque overlay assays were performed as previously described [19]. Briefly, prepared phage cultures were serially diluted in SM buffer so that plated volumes would result in 30–300 plaques per plate to make counting possible. Luria-Bertani (LB) soft agar (0.7%) was heated to boiling and 5 mL aliquots were maintained at a temperature of 50ºC until use. For plating, the soft agar tube was cooled to touch, 250 µL of an overnight (~15hrs) culture of *E. coli*, strain B was added along with either 10 µL or 100 µL of diluted phage. Tube contents were immediately added to the top of an LB plate with gentle swirling to ensure even coverage and mixing without bubbles. Plates were allowed to solidify completely before being inverted and incubated at 37ºC overnight to allow for bacterial lawn growth and plaque formation. The following day, plaques were assessed and the plate dilutions with plaque numbers in the desired range were counted. Plaque counts and dilution were used to determine the amount of phage in the original stock solution. Plating was performed in triplicate with bacteria alone as a negative control.

## Data analysis and graphing

For analyzing run data and application of the macro, Microsoft Excel software was used. To generate the graphs and perform analysis of the accuracy of the macro, GraphPad PRISM software (GraphPad, Boston, MA) was employed.

## Results

### Detection of bacteriophage with a particle size analyzer plus PIDS

Derivations of particle size analyzers have been commonly used in a variety of chemistry-based applications for the quality analysis (QC/QA) of manufactured emulsions, suspensions, and dry powders (e.g., [20,21]). The inclusion of PIDS technology means that particle detection by the PSA now extends to particles of the smallest size (10nm-500nm), the size range bacteriophages fall into. With this technological advance, we hypothesized that detection of nanometer scale biological particles, such as bacteriophages, would be possible with the PSA.

To test this hypothesis, 500 µL of T4 phage was added to the sample port of the PSA and analyzed using PIDS with the parameters described in Methods section 2.2. After three runs, the software provided a differential volume (%) histogram depiction of particle size distribution detected within the solution (Fig 2A) with a dominant peak occurring around 55 nm (Fig 2B, Mean (µm)) indicating that use of the PSA with PIDS would allow for detection of bacteriophages in solution.

### Generation of macro for phage quantification

The PSA is typically used for characterization of size and distribution of particles within a sample, not quantitation of those particles. While we knew we could detect phage using the PSA, true utility of this instrument in phage research would require the ability to quantitate phage particles within a sample. To do this, a computational macro in Excel was developed by Beckman Coulter to allow for input of data from the PSA to be converted into a phage/mL output. Development of a macro was completed by comparing counts of 300nm latex beads from a particle counter, in this case the Multisizer 4e, with results from the PSA. The Multisizer 4e is a particle counter which uses the Coulter principle [22] to quantitate particle number in samples but it cannot accurately detect or quantitate particles below a 200 nm cut off, meaning that it does not have the same capacity as the PSA to detect phages. Several aliquots of different volumes from the same sample of 300 nm latex beads in electrolyte solution were run side by side in the PSA and the Multisizer 4e to establish an algorithm for

# Individual Phages

**T4**  **ΦX**  **M13**

**Purify phage and exchange to SM Buffer**

- Select desired phage program
- Load 0.5-3mL of sample
- Gently mix fluid with transfer pipet
- Perform 2-3 consecutive reads; 4 runs each
- 120 sec read time; 10% pump speed

**Quantitate phages**

**With PhageFOTO**  OR  **With plaque assay**

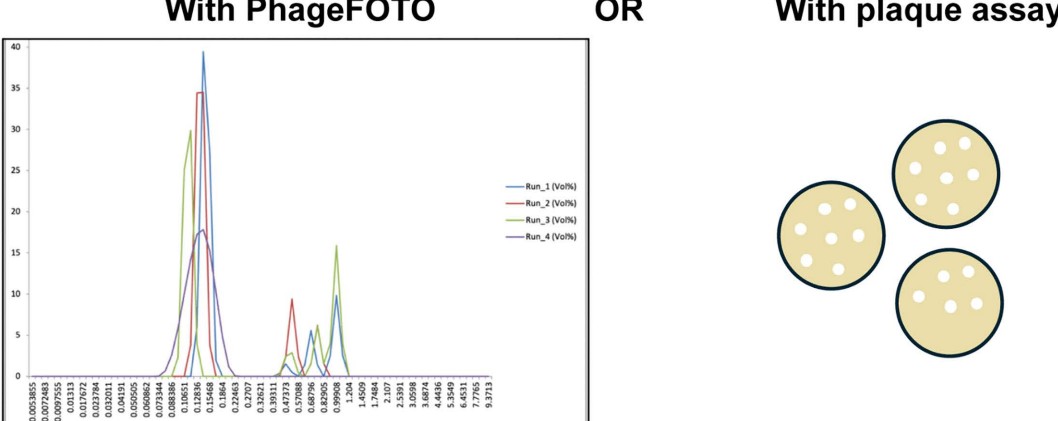

**Fig 1. Phage detection and quantitation.** Workflow of phage type and processing with PhageFOTO and plaque overlay assay to validate use of the PSA with PIDS for phage quantitation.

## A.

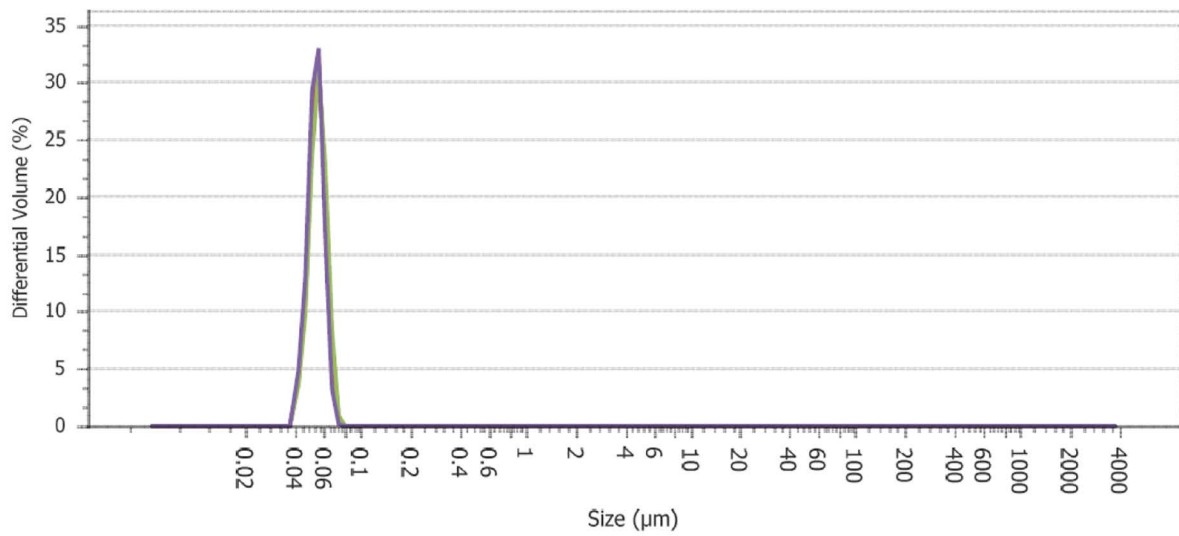

## B.

| Run | 1mL | 1mL | 1mL | 1mL | Avg | CV (%) |
|---|---|---|---|---|---|---|
| | 1 | 2 | 3 | 4 | | |
| D10 (µm) | 0.04647 | 0.04622 | 0.04681 | 0.04565 | 0.04629 | 1.056 |
| D50 (µm) | 0.05472 | 0.05438 | 0.05515 | 0.05341 | 0.05442 | 1.361 |
| D90 (µm) | 0.06323 | 0.06287 | 0.06361 | 0.06153 | 0.06281 | 1.441 |
| Mean (µm) | 0.05484 | 0.05448 | 0.05527 | 0.05344 | 0.05451 | 1.434 |
| Mode (µm) | 0.05544 | 0.05544 | 0.05544 | 0.05544 | 0.05544 | 0.0000 |
| StDev (µm) | 0.006314 | 0.006201 | 0.006422 | 0.005741 | 0.006170 | 4.856 |
| Total (%) | 100.0 | 100.0 | 100.0 | 100.0 | 100.0 | 0.0000 |

**Fig 2. Detection of T4 on PSA.** Sample of T4 run on the PSA with 4 reads resulted in (A) detection of a single differential volume (%) peak with mean particle size distribution at approximately 55 nm. **(B)** Statistics accompanying run of T4 with breakdown of D10, D50, D90, mean, mode, standard deviation (StDev), and total (%) of sample account for each run. Column values for each run are denoted with different color matching the line color associated in histogram **(A)**. Average and CV(%) for all runs are provided on the far right of the statistical table **(B)**.

quantitating beads/mL using the PSA. The subsequent algorithm, which was eventually embedded into the PhageFOTO macro, calculated number of beads/mL using data from the PSA with PIDS and matching it to the same sample counts acquired on the Multisizer 4e (Tables 1 and 2).

The necessary data from the PSA to be input into this macro included the volume percent (Vol%) data, Particle size (µm)/volume distribution, the PIDS percent (PIDS (%)) obscuration, the total volume (in µL) of the PSA sample reservoir (130mL), and the sample volume (in mL) (Fig 3A and 3B; green highlighted fields in S1 File). The equation for the algorithm was as follows:

Vol% x (Particle size (µm)/ Volume) x PIDS (%) obscuration x (volume of sample reservoir (µL)/4/sample volume (mL)) where the "4" is the correction factor necessary to match data from the PSA to the Multisizer 4e.

**Table 1. Comparison of beads/mL values obtained with the algorithm from PSA data as compared to the actual counts obtained with the Multisizer 4e.**

| Volume of Bead Solution Added | Multisizer 4e (Beads/mL) | PSA (Beads/mL) |
|---|---|---|
| 50 µL | $1.17 \times 10^{10}$ $1.28 \times 10^{10}$ | $1.29 \times 10^{10}$ $1.27 \times 10^{10}$ |
| 100 µL | $1.18 \times 10^{10}$ $1.31 \times 10^{10}$ | $1.28 \times 10^{10}$ $1.25 \times 10^{10}$ |
| 150 µL | $1.21 \times 10^{10}$ $1.23 \times 10^{10}$ | $1.20 \times 10^{10}$ $1.19 \times 10^{10}$ |
| 200 µL | $1.15 \times 10^{10}$ $1.01 \times 10^{10}$ | $6.49 \times 10^{10}$ $1.11 \times 10^{10}$ |
| 250 µL | $1.38 \times 10^{10}$ $1.08 \times 10^{10}$ | $1.02 \times 10^{10}$ $1.02 \times 10^{10}$ |

**Table 2. Comparison of PSA and Multisizer 4e beads/mL quantitation of 300nm beads.**

| | Overall PSA | Overall Multisizer 4e | Average | SD | C.V. |
|---|---|---|---|---|---|
| Average | $1.18 \times 10^{10}$ | $1.2 \times 10^{10}$ | $1.19 \times 10^{10}$ | $5.7 \times 10^{7}$ | 0.48% |
| C.V. | 8.36% | 9% | $8.68 \times 10^{-2}$ | $2.25 \times 10^{-3}$ | 2.59% |
| SD | $9.89 \times 10^{8}$ | $1.08 \times 10^{9}$ | $1.03 \times 10^{9}$ | $3.09 \times 10^{7}$ | 2.99% |
| Maximum | $1.29 \times 10^{10}$ | $1.38 \times 10^{10}$ | $1.33 \times 10^{10}$ | $3.05 \times 10^{8}$ | 2.29% |
| Minimum | $1.02 \times 10^{10}$ | $1.01 \times 10^{10}$ | $1.01 \times 10^{10}$ | $6.43 \times 10^{7}$ | 0.63% |

## Quantification of bacteriophages

With the macro in place, we sought to validate use of the PSA to quantify phages of different sizes and shapes accurately. As described in Methods (Phage Detection and Quantitation) and Fig 1, we acquired data from samples containing one of three physiologically different phages (T4- tailed, prolate icosahedron [23], M13- filamentous [24], and ΦX- non-tailed, small icosahedron [25]; See Fig 1 for shapes). The total runs for each of the phages tested from which data was analyzed included 68 runs for T4, 56 runs for ΦX, and 60 runs for M13. Concurrent plaque overlay assays were performed on these same samples to provide a benchmark number with which to compare the phage/mL results generated by the macro using the PSA data for each phage.

Analysis of particle distributions across all runs for individual phages tested found that mean peak sizes for each phage fell into a patterned distribution (Fig 4). T4 peaks were found in ranges from 10–60nm (22 runs), 70–150nm (42 runs), or >150nm (4 runs) (Fig 4A). ΦX peaks were found in ranges from 15–20nm (1 run), 35–90nm (39 runs), and >100nm (16 runs) (Fig 4B). M13 peaks were found in ranges from 10–30nm (10 runs), 45–70nm (45 runs), and >100nm (5 runs) (Fig 4C). That each phage resulted in a distribution of peak sizes suggested a variety of factors influenced peak outcomes. It has been shown that the non-spherical nature of these phages influenced peak shifts due to diffraction of the light occurring differently depending on the plane it hits [26]. Larger peak sizes were likely indication of phage propensity to aggregate and smaller than expected peaks could have been due to breakage of phage structure associated with sample handling. Detection peaks were only found in samples containing phages as application of either SM buffer alone or the flow through fraction from phage concentration resulted in no particles detected by the PSA.

To quantitate phages/mL for all runs of each phage tested, PSA data for each run was input into the macro generated from comparison of 300nm beads on the PSA and the Multisizer 4e (as described above). The phages/mL numbers generated by the original macro were compared to the comparable phages/mL numbers generated by the corresponding

## A.

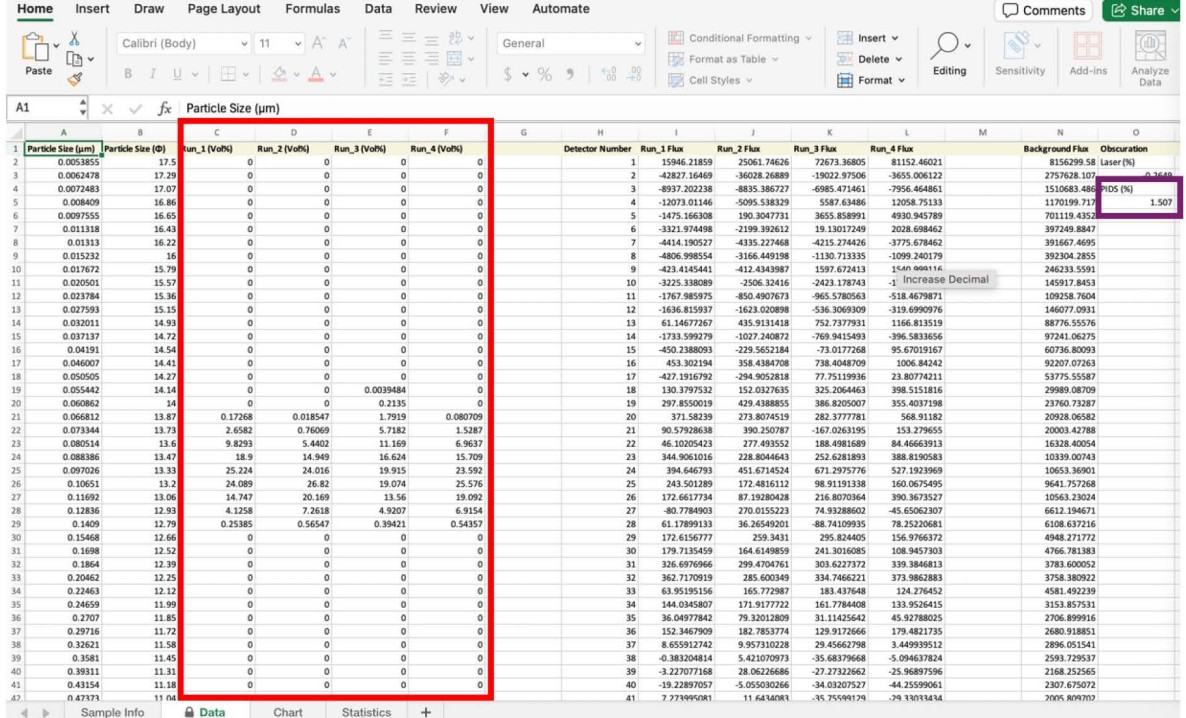

## B.

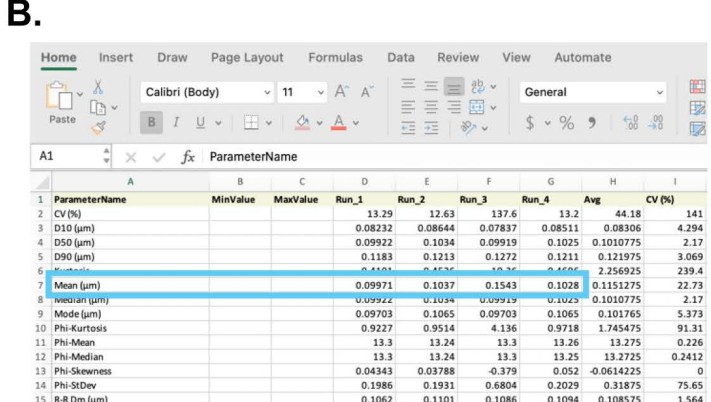

## C.

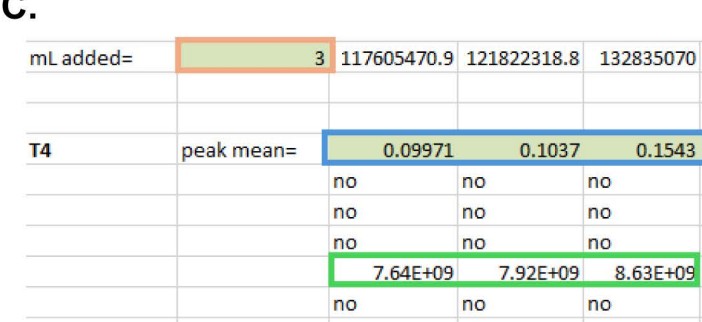

**Fig 3. Elements from PSA data for inclusion into PhageFOTO macro for phages/mL quantitation.** PhageFOTO macro input data required from the PSA exported Excel file for each read includes (A) the volume percent (vol%) data for each run (large red rectangle) and the PIDS (%) Obscuration data (small purple rectangle) found within the "Data" tab and (B) mean (µm) particle size for each run (blue rectangle) found within the "Statistics" tab. **(C)** Within the PhageFOTO macro, sample volume (in mL) is input (orange rectangle) along with mean particle size (µm) for each run (blue rectangle) in a phage specific section (T4, ΦX, M13, or mixed; see S1 File). Calculation of phages/mL is found below mean particle size as the first numerical value displayed (green rectangle). This value is reported as phages/mL.

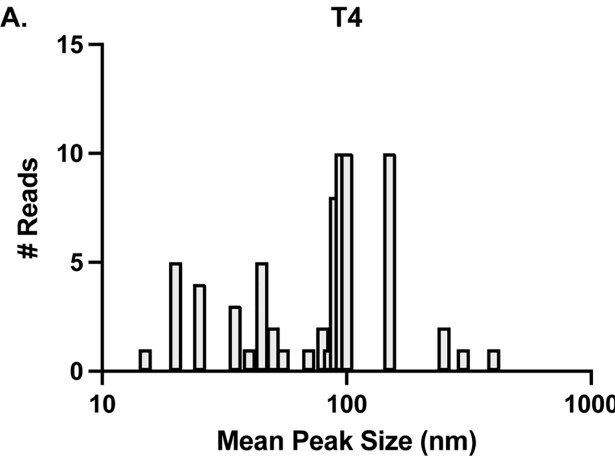

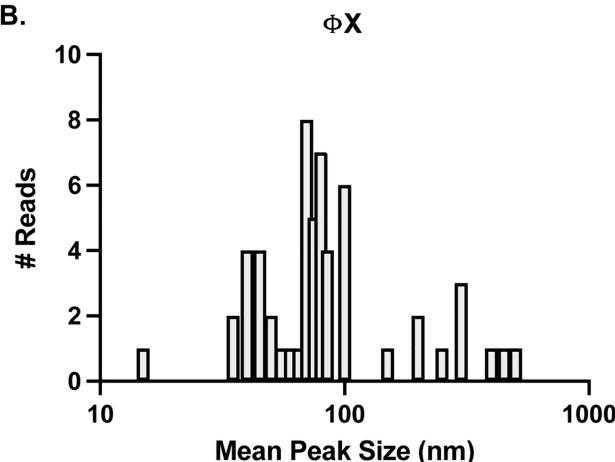

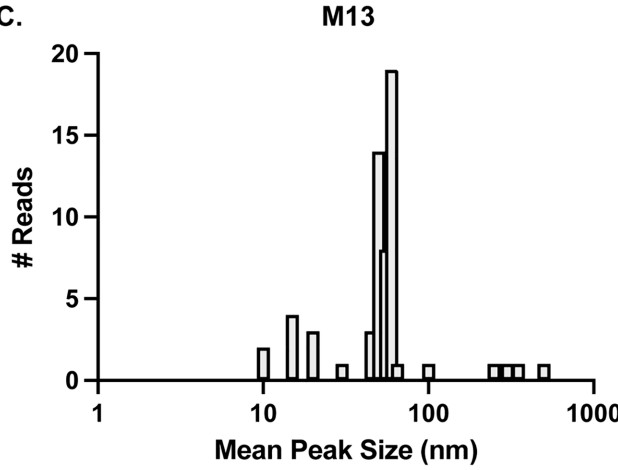

**Fig 4. Distribution of mean particle size on PSA.** Histogram representation of the mean peak sizes of particle distribution for each run as a function of the number of runs for **(A)** T4, **(B)** ΦX, and **(C)** M13. Each column width represents a 5 nm range with the X-axis on a logarithmic scale.

plaque overlay assays. Results showed that PSA numbers were off by one or several logs and that this variance depended on mean particle size. To compensate for differences seen in the macro and the actual phage counts from plaque assay, the mean particle size for each run was then included as part of the calculation and a multiplicative factor specific to each phage was used to allow for close approximation of PSA data to actual phage counts thus creating the final version of the PhageFOTO macro (S1 File). In PhageFOTO, mean peak size from PSA data was copied into the section for the specific phage tested (i.e., "T4, peak mean" in Fig 3C) along with the volume of sample ("mL added", Fig 3C) allowing for enumeration of phages based on the parameters determined for best fit. A series of outputs for each run was shown beneath the mean peak size and the numerical output that came first represented the enumerative calculation for that specific run (Fig 3C).

Once the PhageFOTO macro was created, we determined the number of times it accurately calculated phages/mL as compared to the corresponding plaque assays (± 0.5 log) (S2 File; T4, M13, and PhiX tabs). The accuracy of PhageFOTO for each phage are shown (Fig 5) where the number of correct predictions (connected points, right y-axis) were overlaid with the histogram distribution of the number of runs (columns, left y-axis) that fell within any given mean peak size (nm, x-axis) for each phage tested. From these comparisons, we found that PhageFOTO was highly accurate for quantitation (phages/mL) of all individual phages tested as compared to the plaque overlay assay. Results were 94 ± 2.7% accurate for T4 (Fig 5A), 89 ± 4.3% accurate for PhiX (Fig 5B) and 98 ± 1.7% accurate for M13 (Fig 5C).

### Limit of detection of particle size analyzer with PIDS

The lower limit of detection of the PSA is ~10nm due to the incorporated PIDS system. However, there were several parameters that could be adjusted to allow for higher sensitivity towards smaller particles. These included using low speed within the sampling chamber (the small size of the particles means that they won't settle within the sampling chamber over typical measuring times), using gentle pipetting within the sample chamber after sample addition to ensure particles are distributed across the imaging window, increased read times, and allowing for more reads per sample. All these protocol modifiers worked to increase the amount of time a sample within the detection window was analyzed. To characterize the particle concentration limit of detection using our defined instrument set-up, a stock solution of CsCl gradient purified T4 phages of $10^9$ phages/mL was diluted with a 10-fold serial dilution into SM Buffer. Counter settings were maintained at 10% pump speed, 120 second read time, and 3 reads for each sample. Samples from each dilution were loaded into the counter sample port starting with a 0.25 mL volume and increasing by 0.25 mL increments up to a 1 mL final volume depending on whether the PSA detected anything. We found upon several repetitions with these serial dilutions that counter detection of $10^8$ and $10^9$ phages/mL occurred consistently with 0.25–0.5 mL of sample. Dilution to $10^7$ phages/mL required higher sample volumes of 0.75–1 mL, and dilutions with <$10^7$ phages/mL were undetectable by the PSA regardless of volume tested (up to 5 mL).

### Discussion

In order to perform bacteriophage research, it is essential to know the number of phage particles that exist within a sample. However, inherent challenges have included their small size, challenging purification, and lack of a consensus sequence such as 16S or 18S. To this end, several different methods have been developed to address the enumeration of phages within samples [2,3,6]. For instances where phages and their bacterial hosts are known and understood, enumeration is easier with the plaquing method or qPCR with phage specific primers. For samples that are more challenging (i.e., environmental isolates), labeling and examination by microscopy (TEM or EFM) or flow cytometry have been employed. These methods are time consuming, can require expensive instrumentation and expertise, and have been shown to have variable reliability dependent on sample handling procedures [6]. Thus, development of a rapid, consistent, simple method for quantitation of phages in solution is desirable. Herein, we described novel use of a particle size analyzer with PIDS detector and fluidic ULM module (comparable in cost to a flow cytometer or confocal microscope) for rapid quantitation of

**A.**

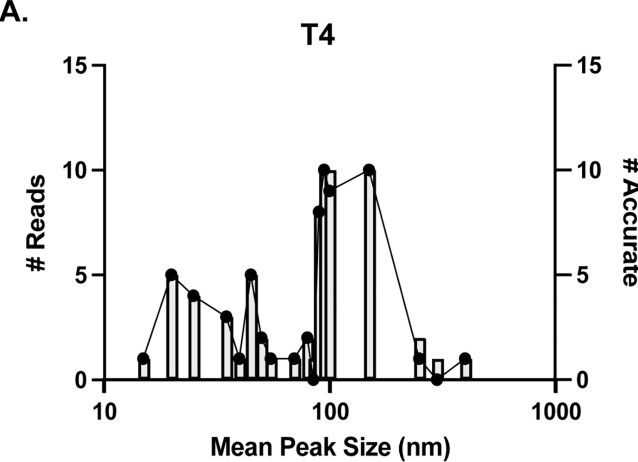

**B.**

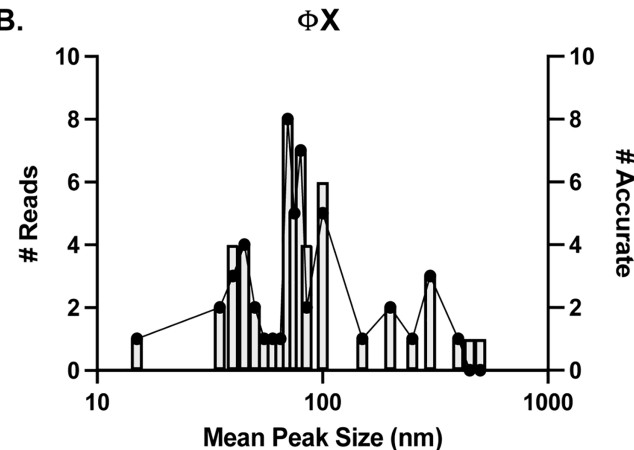

**C.**

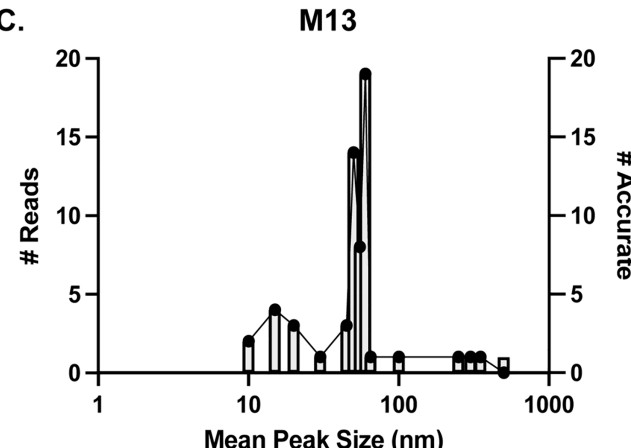

**Fig 5. Validation of accuracy of PhageFOTO macro for quantitation.** Number of correct phages/mL predictions by the PhageFOTO macro (connected dots, right axis) for each series of runs across the distribution of mean peak sizes (gray columns, left axis) for **(A)** T4, **(B)** ΦX, and **(C)** M13. Each column width represents a 5 nm range with the X-axis on a logarithmic scale.

phages without need for fluorescent labeling or prior knowledge of bacterial host. The combination of the principles outlined by the Mie scattering theory for spherical particles [17] with light scattering data acquired by the PIDS detectors from three different wavelengths and two polarizations allowed for definition of the size distribution of particles in the submicron range down to 10 nm, ideal for phage detection. Examination of phage size distributions in individual phage preps demonstrated a frequency distribution with some peak sizes more frequently represented in a phage-specific way. For example, the most frequent peak sizes were found in the 90–155 nm range for T4, 70–90 nm range for ΦX, and 50–65 nm range for M13. In an ideal situation, phage solutions would contain uniformly monodisperse phages resulting in generation of a consistent, single peak associated with the expected particle size for a single phage. However, only for T4 do the most frequent peak sizes correspond with what would be expected for an individual phage particle, and even for T4 samples there is a broad distribution. This distribution was partially due to the non-spherical nature of phages [26] and their heterogeneity and polydispersity within any given sample. In fact, several environmental factors contribute to phage aggregation [27,28] and M13 and ΦX were more likely to aggregate as compared to T4 resulting in scattering patterns consistent with a mean peak size larger than would be expected for individual M13 or ΦX phages. Additionally, while ΦX structure was the closest to the spherical particles that Mie scattering was based on, filamentous M13 was the furthest from this structure meaning that its scattering properties were farthest from the models fitting spherical particles, and thus farther from conforming to an expected mean peak size associated with the PSA software.

The physical shape and density of phages (aggregated versus monodisperse) also influenced the likelihood that the PhageFOTO macro would accurately quantitate the number of phages/mL [26]. As more data is collected for individual phages of all types across the spectrum of mean peak sizes, the PhageFOTO macro can be adjusted to ensure that enumeration is as accurate as possible. All data has been provided in supplementary info to aid the user in determining best fit for their needs and to outline how data on other phage types should be collected (S2 File). The particle detection limit for this method was on par with other methods such as EFM and flow cytometry where detection of $10^7$–$10^9$ viral like particles/mL is possible [6]. Next steps will include not only working to increase sensitivity of this detection limit but to collect data from mixed phage solutions to begin building models that will allow for use of this instrumentation in enumeration of phages from environmental samples. In this manner, PhageFOTO provides a rapid and reproducible method to quantitate phages from any phage-containing sample.

## Supporting information

**S1 File. PhageFOTO Excel macro.** Excel macro for determining phage counts/mL. Green highlighted regions indicate areas where data from PSA is input. "Run(vol%)" data from all runs is inserted on the left, "PIDS(%)" from the data is input in I5, the peak means obtained from the statistics tab in the data file is input for the phage type tested, and the volume of sample added is input in mL format in cell I138. Counts are generated in the cells underneath the input peak mean values with the lowest count representing the counts/mL.
(XLSX)

**S2 File. Phage counts obtained from PSA compared to plaque overlay.** Counts/mL for each phage tested (T4, M13, and PhiX) from either the plaque assay ("PFU" column) or the PSA ("Macro" column). Column D is mean peak size found for each sample of each type of phage, and Column E denotes whether PSA finding fall within 0.5 log of counts determined with plaque assay (Y = yes, N = No).
(XLSX)

## Acknowledgments

The Edgar Martinez of Beckman Coulter particle counter technical support team provided information on technical specifications of instrument, Multisizer 4e vs LSXR data, the original macro for counting phages, and follow on support for analysis of instrument data.

## Author contributions

**Conceptualization:** Amanda Carroll-Portillo, Henry C Lin.

**Data curation:** Amanda Carroll-Portillo, Cody A Braun.

**Formal analysis:** Amanda Carroll-Portillo.

**Funding acquisition:** Henry C Lin.

**Methodology:** Amanda Carroll-Portillo, Cody A Braun, Edgar Martinez.

**Software:** Amanda Carroll-Portillo, Edgar Martinez.

**Supervision:** Amanda Carroll-Portillo.

**Validation:** Amanda Carroll-Portillo, Edgar Martinez.

**Writing – original draft:** Amanda Carroll-Portillo.

**Writing – review & editing:** Amanda Carroll-Portillo, Henry C Lin.

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
