## [Decision Letter · Decision Letter 0]

15 Dec 2025

Dear Dr. Carroll-Portillo,

Thank you for submitting your manuscript to PLOS ONE. After careful consideration, we feel that it has merit but does not fully meet PLOS ONE’s publication criteria as it currently stands. Therefore, we invite you to submit a revised version of the manuscript that addresses the points raised during the review process.

**Dear respected Authors, Based on the reports that we received from our reviewers, I have decided that your manuscript needs a major revision. Please follow the recommendations raised by evaluators and resubmit your revised manuscript. Best regards,**

We look forward to receiving your revised manuscript.

Kind regards,

Mohammad Faezi Ghasemi, Ph.D

Academic Editor

PLOS One

Journal Requirements:

“This work was supported by the Winkler Bacterial Overgrowth Research Fund (BRINM 217)”

“the Winkler Bacterial Overgrowth Research Fund (BRINM 217)”

“the Winkler Bacterial Overgrowth Research Fund (BRINM 217)”

Reviewers' comments:

Reviewer's Responses to Questions

**Comments to the Author**

1. Is the manuscript technically sound, and do the data support the conclusions?

Reviewer #1: Yes

Reviewer #2: Yes

2. Has the statistical analysis been performed appropriately and rigorously?

Reviewer #1: Yes

Reviewer #2: N/A

3. Have the authors made all data underlying the findings in their manuscript fully available?

Reviewer #1: Yes

Reviewer #2: Yes

4. Is the manuscript presented in an intelligible fashion and written in standard English?

Reviewer #1: Yes

Reviewer #2: Yes

Reviewer #1: - To enhance the informativeness of the abstract, consider incorporating key performance metrics such as the method's detection limit and the quantitative accuracy such as percent recovery achieved for each phage type.

- The specificity of the assay requires further validation. The inclusion of negative controls (phage-free lysate or buffer-only samples) is necessary to demonstrate that background signals do not significantly contribute to the reported peaks.

- Please expand the Discussion to provide a quantitative analysis of how deviations from ideal sphericity impact particle size estimation and the subsequent accuracy of phage enumeration using the PhageFOTO macro.

- The Discussion would benefit from a more direct comparative analysis, situating PhageFOTO among current rapid methods (specific flow cytometry or microscopy protocols). A comparison focusing on practical parameters like cost, hands-on time, required expertise, and sample volume would be valuable.

-

Reviewer #2: This study introduces a novel application of a particle size analyzer with PIDS technology (PhageFOTO) for rapid quantification of bacteriophages. The core idea is innovative and addresses a genuine need for faster, label-free phage enumeration. The method is validated against the plaque assay using three distinct phages, demonstrating promising accuracy. However, the manuscript in its current form requires revision to improve clarity, methodological rigor, and scholarly presentation before it is suitable for publication.

**Do you want your identity to be public for this peer review?** For information about this choice, including consent withdrawal, please see our Privacy Policy

Reviewer #1: No

Reviewer #2: No

---

## [Author Response · Author response to Decision Letter 1]

19 Jan 2026

The authors would like to thank the reviewers for taking the time and effort necessary to critique this article and provide suggestions for its betterment. Please find our responses to comments and suggestions below.

Reviewer 1:

1. The abstract lacks quantitative results; it does not provide numerical values, particle size ranges, or statistical indicators. Additionally, the final sentence fails to clearly state the main conclusion or its implications.

We have revised the abstract to include the range of phage sizes tested and statistics for PSA measurements. We have also stated our main conclusion and implications more clearly. See lines 20-27

2. Inconsistent use of verb tenses was observed. The entire abstract should be rewritten in the past simple tense. For example, in line 23, "has expanded" (present perfect) and in line 33, "We demonstrate" (present simple) should both be converted to the past simple tense.

Verb tenses are now consistent.

3. The limit of detection is not mentioned in the abstract.

The limit of detection has been added to the abstract. See lines 24-25

4. The literature review is fragmented and lacks logical coherence. Several references are outdated and should be replaced with more recent studies published after 2019. For example, references cited on lines 392 (2000) and 395 (2009) are outdated. It is recommended to include more recent systematic reviews published after 2020.

There are limited systematic reviews on phage quantitation methods. We have added more recent references relating to flow cytometry detection of phages.

5. The explanation of fluorescent microscopy methods in lines 64–75 is somewhat lengthy and could be more concise.

We have rewritten this section to be more succinct. See lines 53-57.

6. The scientific knowledge gap is not clearly defined. The novelty and specific contribution of the study remain unclear. Although lines 87–96 mention the limitations of existing methods, they do not explicitly state the specific problem that this new method addresses, which others cannot solve.

We now state that this method uses a rapid, label-free method that does not require knowledge of the bacterial host (Lines 92-95).

7. It should be explicitly stated in lines 96–107 that this is the first application of PSA combined with PIDS for phage counting.

We have done this (Line 92)

8. The sample preparation procedures are inadequately described. Key details such as initial concentrations, incubation times, and temperatures are missing. Although lines 136–139 mention an incubation time of 3–4 hours at 37°C, the initial concentrations of bacteria or phages are not provided.

More information has been added. See throughout methods section.

9. Line 141: The text mentions a µm syringe filter, but it does not specify the brand or the material of the filter.

The filter material has been added (PTFE). Several brands have been used with all being equally effective, as such no one brand is listed.

10. Critical details regarding the Amicon Ultra protocol, including membrane cutoff, centrifugation speed, duration, and washing steps, are missing, severely limiting reproducibility. While the cutoff (10 kDa MWCO) and speed (4000 rpm) are specified, the exact centrifugation time—stated only as volume is reduced to 200–500 µL vague. Additionally, washing or equilibration steps are not described. The centrifugation temperature should also be provided.

Additional information has been added to the protocol where necessary. Speed, temperature, duration, and cutoff have all been included. The time of centrifugation has been put in but is a suggested starting point as it can be insufficient for generating the final 200-500ul volume. Getting to that final volume is the more important point. Washing and equilibration steps are not performed, just centrifugation to concentration. See lines 119-123.

11. The exact volumes of the bacterial suspension and phage dilution are unclear, and the order of mixing is not specified. Line 171 mentions the bacterial volume (250 µL) and phage volume (10 or 100 µL), but the bacterial concentration (OD600) is not reported. Although the mixing order is implied, it could be stated more explicitly.

Text states using overnight cultures of bacteria with added information of how long growth was allowed. OD600 is variable in overnight cultures but as long as there is turbid growth, it is sufficient for this protocol to work. Aiming for a specific OD600 is not necessary if using an overnight culture. Mixing order has been clarified further. See lines 143-157.

12. Line 175: “Plates were allowed to solidify completely,” but the solidification time is not specified.

No solidification time is listed as it varies by environmental conditions. The important thing is to have it completely solidified prior to inversion for incubation overnight.

13. No information is provided regarding the number of replicates or the use of positive and negative controls. Line 154 mentions “in triplicate three separate times,” but negative controls (e.g., blank buffer) and positive controls (e.g., standard particles) are not clearly described.

Description of controls has been added where necessary. In methods, use of bacteria alone is described as the negative control for plaquing assays (Line 161). Use of SM Buffer alone has been described as the negative control for PSA experiments (Lines 249-251).

14. The Particle size distribution results are presented without adequate textual interpretation or clear connection to the main research objective. Although lines 263–267 report peak ranges, the interpretation of multi-peak phenomena (e.g., aggregation) is not thoroughly addressed in the Results section and is instead deferred to the Discussion.

We have added the observation and associated reference to explain the peak distribution to the results section (lines 270-272) and have left the more detailed description in the discussion section.

15. The discussion primarily reiterates the results instead of offering interpretation. Comparisons with previous studies are limited. Although lines 334–353 explain the reasons for size distributions, direct comparisons with other counting methods (e.g., flow cytometry) are minimal.

We have expanded in the discussion to include further discussion on interpretation, comparison with other counting methods, and instrumentation cost.

16. The Practical limitations of the method, such as the cost of the instruments, should be emphasized more in the Discussion section.

We have included the instrument cost relative to instrumentation for other methods. See line 322.

17. Numerous grammatical errors, overly long sentences, and inconsistent tense usage are present. A major revision of the English language is required.

We have reviewed with focus on clarification of the language and matching verb tense.

18. Line 64: “Sybr” should be “SYBR.”

This has been changed. See line 54.

19. Line 148: The term “CsCl” is used without prior explanation; it should be introduced as cesium chloride (CsCl) when first mentioned to ensure clarity.

CsCl has been defined at its first use in line 126

20. Line 306: “We needed to understand” should be changed to “We needed to understand” in the past simple tense.

We removed this

21. Long sentences, such as those in lines 73–75, could be broken up for better readability.

We have reviewed and changed sentence structure to be succinct

22. In the References section (lines 390–460), there are formatting inconsistencies, such as irregular capitalization and punctuation.

References were input using EndNote Vancouver style of citation

Reviewer #2: This study introduces a novel application of a particle size analyzer with PIDS technology (PhageFOTO) for rapid quantification of bacteriophages. The core idea is innovative and addresses a genuine need for faster, label-free phage enumeration. The method is validated against the plaque assay using three distinct phages, demonstrating promising accuracy. However, the manuscript in its current form requires revision to improve clarity, methodological rigor, and scholarly presentation before it is suitable for publication.

We thank this reviewer for their input. We have edited this article substantially to increase clarity, better demonstrate our methodological rigor, and improve presentation of our results and discussion.

---

## [Decision Letter · Decision Letter 1]

3 Feb 2026

Rapid bacteriophage quantification with a particle size analyzer combined with Polarization Intensity Differential Scattering (PIDS) detector

PONE-D-25-61226R1

Dear Dr. Carroll-Portillo,

We’re pleased to inform you that your manuscript has been judged scientifically suitable for publication and will be formally accepted for publication once it meets all outstanding technical requirements.

Kind regards,

Mohammad Faezi Ghasemi, Ph.D

Academic Editor

PLOS One

Additional Editor Comments (optional):

Reviewers' comments:

Reviewer's Responses to Questions

**Comments to the Author**

Reviewer #1: All comments have been addressed

Reviewer #2: (No Response)

2. Is the manuscript technically sound, and do the data support the conclusions?

Reviewer #1: Yes

Reviewer #2: Yes

3. Has the statistical analysis been performed appropriately and rigorously?

Reviewer #1: Yes

Reviewer #2: N/A

4. Have the authors made all data underlying the findings in their manuscript fully available?

Reviewer #1: Yes

Reviewer #2: Yes

5. Is the manuscript presented in an intelligible fashion and written in standard English?

Reviewer #1: Yes

Reviewer #2: No

Reviewer #1: Dear editor,

The comments have been addressed.The findings reported in the Results section are now fully derived from the collected data and performed analyses, with clear consistency established between these two sections. The Discussion section has been significantly expanded and strengthened. The current findings are now comprehensively compared with the existing literature. The Conclusion has been rewritten and focused to concisely and clearly reflect the main objectives of the research, its most significant findings, and suggested directions for future work. The article is acceptable.

Reviewer #2: I. Linguistic and Grammatical Deficiencies

These errors undermine the professionalism and clarity of the manuscript.

The abstract inconsistently uses verb tenses, mixing the present perfect and simple past forms.

• Subject-Verb Agreement Error: A fundamental grammatical mistake persists (Page 11, Line 206): "The necessary data... includes..." Data is a plural noun; it should be "The necessary data... include..."

• Several sentences are awkwardly phrased and unclear; they need to be rewritten to improve clarity.

• Example (Page 4, Lines 57–58): "Errors in EFM arise from inconsistent nucleic acid labeling, fluorescent signals from non-phage particles, or limitations in instrumental resolution.

• Misuse of Articles and Prepositions: The incorrect use of articles such as "the," as well as prepositions, is frequent, resulting in unnatural English. For example, on page 5, line 80, the phrase "moves detection of PSAs into the range" should be revised to "extends the detection range of PSAs.

• Inconsistent Terminology: The method's name is written variably as "PhageFOTO" and "phagefOTO." It should be standardized as "PhageFOTO" throughout. Brand names such as "SYBR Green" require consistent capitalization.

II. Methodological Ambiguities and Omissions

These issues significantly undermine the reproducibility of the study, which is a cornerstone of scientific research.

• Insufficient details provided about the protocol.

• Centrifugation Time (Page 7, Ln 123): The instruction "or until the volume is reduced to 200-500 μL" is vague. It is recommended to specify a precise time range (e.g., "for 15–30 minutes") alongside the volume target to ensure clarity and consistency.

• Bacterial Host Concentration (Page 8, Line 150): The phrase "an overnight culture for plaque assays is imprecise. It is essential to use a standardized metric, such as OD600 or a mid-log phase culture, to ensure reproducibility.

• Filter Specification (Page 6, Line 122): While "0.45 μm PTFE" is specified, the brand or source should be included for critical materials to ensure consistency and reproducibility.

• Inadequate Description of Controls: Although negative controls (e.g., SM buffer alone) are mentioned in the responses, their description in the main Methods section (Page 8) is too brief. A dedicated sentence should explicitly list all controls used for both PSA and plaque assays to ensure clarity and reproducibility.

III. Scientific Limitations and Presentation Flaws

The discussion and results sections should more critically and clearly address the limitations of the method.

• High Detection Limit: The stated limit of detection (~10⁷ phages/mL) (Page 15, Line 310) represents a significant limitation when analyzing dilute environmental samples. This constraint should be explicitly emphasized and compared with other detection methods in the Discussion section.

• Intrinsic Method Limitation – Aggregation and Shape: The results (Page 13, Lines 250–251) correctly note that larger peaks likely indicate phage aggregation. This observation highlights a fundamental limitation of the technique, especially for phages such as M13 and ΦX, which are prone to aggregation. The Discussion section (Page 17) should include a dedicated paragraph explaining how particle aggregation and non-spherical morphology fundamentally challenge Mie theory-based detection and quantification, thereby compromising accuracy.

- Lack of Validation on Complex Samples: The study validates the method exclusively on pure, isolated phage preparations. The authors' mention of future work involving "mixed phage solutions" (Page 17, Line 357) implicitly acknowledges this significant limitation for environmental applications. This gap should be explicitly stated as a current limitation in the manuscript's conclusion.

Calibration Dependency: The requirement for a "multiplicative factor specific to each phage" (Page 13, Line 267) indicates that the macro is not a universal calculator. Each new phage type must be calibrated against a gold standard. This crucial aspect of the method's reliance on prior calibration should be emphasized in the Discussion section rather than being buried in the Results.

Repetitive and Descriptive Discussion: The Discussion section (pages 16–17) excessively reiterates the results (e.g., explanations for peak size distribution) instead of offering a thorough interpretation of their implications. It should provide a more comprehensive comparison with existing methods (e.g., flow cytometry, EFM) and address practical limitations of the method, such as instrument cost.

IV. Structural and Stylistic Recommendations

Streamline the introduction by making the literature review on existing methods (e.g., EFM) more concise to better emphasize the identified research gap.

- Improve Figure and Table Integration: Actively refer to figures and tables within the results narrative (e.g., "As shown in Figure 2A...").

The manuscript requires a thorough edit by a native English speaker with expertise in scientific writing to correct all grammatical errors, improve sentence flow, and ensure a formal academic tone.

**Do you want your identity to be public for this peer review?** For information about this choice, including consent withdrawal, please see our Privacy Policy

Reviewer #1: **Yes:** Ashraf Kariminik

Reviewer #2: No

---

## [Editor Report · Acceptance letter]

PONE-D-25-61226R1

PLOS One

Dear Dr. Carroll-Portillo,

I'm pleased to inform you that your manuscript has been deemed suitable for publication in PLOS One. Congratulations! Your manuscript is now being handed over to our production team.

Kind regards,

on behalf of

Dr. Mohammad Faezi Ghasemi

Academic Editor

PLOS One